Initial decomposition of floating leaf blades of waterlilies: causes, damage types and impacts

Klok Peter F. 1 2
van der Velde Gerard g.vandervelde@science.ru.nl 1 3
1 Department of Animal Ecology and Physiology, Institute for Water and Wetland Research, Radboud University , Nijmegen , Netherlands
2 Department of Particle Physics, Institute for Mathematics, Astrophysics and Particle Physics, Radboud University , Nijmegen , Netherlands
3 Naturalis Biodiversity Center , Leiden , Netherlands
Gessner Mark
Electronic publication date: 2019 Jun 21
Publication date: 2019
Volume: 7
Electronic Location ID: e7158
Received 2018 Jul 5; Accepted 2019 May 18
Copyright: ©2019 Klok and van der Velde
Copyright year: 2019
Copyright holder: Klok and van der Velde
License: This is an open access article distributed under the terms of the Creative Commons Attribution License, which permits unrestricted use, distribution, reproduction and adaptation in any medium and for any purpose provided that it is properly attributed. For attribution, the original author(s), title, publication source (PeerJ) and either DOI or URL of the article must be cited.
License URL: https://creativecommons.org/licenses/by/4.0/

Keywords: Decomposition causes, Floating leaf blade decomposition, Fresh water body, Nymphaeaceae, Nymphaeid growth form, Seasonal change

Funding: The authors received no funding for this work.

==============================
The initial decomposition of large floating-leaved macrophytes, such as waterlilies, can be studied by following changes in leaf damage and area loss of leaf blades tagged in their natural environment. This approach was taken in the present study to examine the initial decomposition patterns of floating leaf blades of Nuphar lutea (L.) Sm., Nymphaea alba L. and Nymphaea candida C. Presl at three freshwater sites differing in nutrient status, alkalinity and pH. Floating leaf blades of the three plant species were tagged and numbered within established replicate plots and the leaf length, percentages and types of damage and decay of all tagged leaves were recorded weekly during the growing season. Microbial decay, infection by phytopathogenic fungi (Colletotrichum nymphaeae) and oomycetes (Pythium sp.), consumption by pond snails, and mechanical factors were the most important causes of leaf damage. Several types of succession comprising different causes of damage were distinguished during the season. For example, young floating leaves are affected by more or less specialized invertebrate species consuming leaf tissue, followed by non-specialized invertebrate species feeding on the damaged floating leaves. In the two investigated hardwater lakes the seasonal patterns of initial decomposition differed between Nymphaea and Nuphar.

Introduction

The decomposition of leaf blades of floating-leaved macrophytes commences when the leaves are still connected to the parent plant. The usual approach to study this process is to place detached or harvested plant material in litter bags (Brock et al., 1982; Wieder & Lang, 1982; Taketani et al., 2018). Much less attention has been paid to the initial decomposition of aquatic macrophytes before detachment or harvesting. Decomposition in these natural conditions involves a complex set of interacting processes, which can be classified into internal (physiological) and external (abiotic or biotic) processes (Van der Velde et al., 1982). Often, various stages and causes of decomposition occur on one plant or even on a single leaf.

During initial decomposition, macrophyte tissue can be used by herbivores and by phytopathogenic and saprotrophic microorganisms. Before death, the plant tissue senesces and further decomposition and disintegration is initiated by weak pathogens and facultative herbivores, leading to the production of debris and fecal pellets. The chemical composition of plant tissue also changes during senescence due to the hydrolysis of macromolecules, which can weaken tissue structure, the resorption of nutrients like N and P as well as carbon compounds such as starch, and the loss of secondary compounds. Furthermore, leaves are colonized by microorganisms, which make the tissue more attractive for detritivorous macroinvertebrates (Rogers & Breen, 1983).

The phases of initial decomposition can be studied well in floating leaf blades (laminae) of large-leaved plants such as waterlilies (Nymphaeaceae) which exist for a relatively long time, on average 38–48 days, and whose turnover is low (P/Bmax 1.35–2.25 yr−1) (Klok & Van der Velde, 2017). Waterlilies occur worldwide (Conard, 1905; Wiersema, 1987; Padgett, 2007) and in many types of water bodies differing in physico-chemical conditions (Van der Velde, Custers & De Lyon, 1986). Waterlilies typically occupy a fixed position in the plant zonation in the littoral zone of lakes between emergent and submerged macrophytes. The nymphaeid growth form combines floating leaves with rooting in the sediment (Luther, 1983; Den Hartog & Van der Velde, 1988). In addition, waterlilies produce thin underwater leaves and aerial leaves when crowding occurs at the water surface or water levels are lowered (Glück, 1924; Van der Velde, 1980).

Floating leaf blades of waterlilies develop under water and subsequently unroll at the water surface where they are attacked by various organisms, although young leaves can already be attacked under water before they unroll (Lammens & Van der Velde, 1978; Van der Velde et al., 1982; Van der Velde & Van der Heijden, 1985; Martínez & Franceschini, 2018). Responses of waterlilies to attacks include replacing old leaves by new ones, shifting from floating leaves to underwater leaves (Kouki, 1993), producing hydrophobic epicuticular wax layers (Riederer & Müller, 2006; Aragón, Reina-Pinto & Serrano, 2017; Fig. 1), spines (Zhang & Yao, 2018), sclereids containing calcium oxalate crystals (Brock & Van der Velde, 1983; Franceschi & Nakata, 2005), tough tissue (Kok et al., 1992; Mueller & Dearing, 1994), and plant secondary metabolites such as alkaloids (Hutchinson, 1975) and phenolics (Kok et al., 1992; Vergeer & Van der Velde, 1997; Smolders et al., 2000; Martínez & Franceschini, 2018). This means that only specific species are able to attack the fresh plant tissue. These species are more or less specialized and often restricted to particular plant taxa. Other species colonize the leaves at later stages after the defense system has been weakened (Kok et al., 1992). Damage of leaves can induce the leaching of soluble carbohydrates such as oligosaccharides and starch, proteinaceous and phenolic compounds, some of which can be rapidly metabolized by microorganisms (Brock, Boon & Paffen, 1985). Partially decayed floating leaves sink to the bottom, where they provide a resource fuelling detritus-based benthic food webs and continue being decomposed (Brock, 1985; Van der Velde & Van der Heijden, 1985; Kok & Van der Velde, 1991; Kok et al., 1992; Kok, 1993).

Figure 1 Defense system.

A newly enrolled floating leaf of N. alba showing the hydrophobic wax layer as indicated by water droplets.

The present study summarizes causes and patterns of initial decomposition of floating leaves of three species of waterlilies in three water bodies differing in pH, alkalinity, nutrient levels and surrounding land use. Data from previous studies were compiled to answer three questions: (1) what are the causes and patterns of initial decomposition of floating leaves? (2) What is the impact of each cause? (3) How does initial decomposition progress during the season?

Materials and Methods

Sites

Field research took place in dense, nearly mono-specific stands of waterlilies in three different water bodies located in The Netherlands: Haarsteegse Wiel (HW), Oude Waal (OW) and Voorste Goorven (VG) (Table 1). Three plots were laid out in stands of Nuphar lutea (L.) Sm. (HW and OW in 1977; VG in 1988), two plots in stands of Nymphaea alba L. (OW in 1977; VG in 1988) and one plot in a stand of Nymphaea candida C. Presl in J. et C. Presl (HW in 1977). The plots were accessed with a small zodiac, which was navigated by gently paddling. Otherwise no boating or navigation occurred in the water bodies, which prevented damage of the plants by propellers.

Table 1 Characteristics of the three study sites in The Netherlands to investigate the initial decomposition of floating leaf blades of waterlilies.

Characteristic	Haarsteegse Wiel (HW)	Oude Waal (OW)	Voorste Goorven (VG)	
Type of water body	Breakthrough lake	Oxbow lake with three breakthrough ponds	Moorland pond	
Location	51°43′05″N, 5°11′07″E	51°51′13″N, 5°53′35″E	51°33′53″N, 5°12′26″E	
Area (ha)	18	25	5	
Maximum depth	17 m	1.5 m and 6–7 m	2 m	
Water level fluctuations	Low	High in winter and spring	Low	
Stratification	In summer, thermocline at 4–6 m	No	No	
Hydrology	Precipitation, evaporation, groundwater seepage	Precipitation, evaporation, groundwater seepage, river overflow	Precipitation, evaporation, groundwater seepage	
Surrounding vegetation	Trees, shrubs, reeds, grassland	Grassland	Forest	
Wind and wave action	Low	Moderate	Moderate	
Bottom	Sand, organic (sapropel)	Sand, clay, organic (sapropel)	Sand, organic (sapropel)	
Trophic state	Eutrophic	Highly eutrophic	Oligotrophic	
Alkalinity (mmol L−1)a	1.5	4.3–6.7	<0.01–0.07	
pHa	7.1–8.5	6.7–8.3	4.7–5.5	
Sampling year	1977	1977	1988	
Macrophyte species (water depth of plot)	Nuphar lutea (1.5 m), Nymphaea candida (2.5 m)	Nuphar lutea (1.5 m), Nymphaea alba (1.5 m)	Nuphar lutea (2 m), Nymphaea alba (2 m)	
Notes.

a From Brock, Boon & Paffen (1985) and Kok, Velde & Landsbergen (1990).

Haarsteegse Wiel, located in the Province of Noord-Brabant, originates from two connected ponds created by dike bursts along the River Meuse. The now isolated water body is eutrophic and has a relatively low alkalinity. The water level fluctuates, depending on precipitation, groundwater seepage and evaporation. Stratification of the water column occurs during summer. The lake bottom consists of sand and an organic layer with increasing thickness towards the littoral zone. The waterlily beds are situated in the wind-sheltered part of the lake.

Oude Waal in the Province of Gelderland is a highly eutrophic oxbow lake in the forelands of the River Waal. Depth during the growing season is shallow, except for three connected breakthrough ponds. The water level is dependent on precipitation, groundwater seepage, overflow of the River Waal in winter or spring, which strongly influences water chemistry and quality, and evaporation. The bottom consisting of clay and sand is covered by an organic layer of varying thickness in the nymphaeid beds.

Finally, Voorste Goorven in the Province of Noord-Brabant is a shallow, oligotrophic, isolated, culturally acidified moorland pond with very low alkalinity. It is surrounded by forests stocking on poorly buffered sandy soils. The hydrology is mainly dependent on precipitation, groundwater seepage and evaporation.

Leaf area

The potential and actual leaf areas were determined to quantify leaf area loss. The potential area refers to the area of the intact leaf. The actual area was defined as the potential area minus the area that was missing. The potential leaf area was calculated by using a quadratic regression to relate it to leaf length (Van der Velde & Peelen-Bexkens, 1983; Klok & Van der Velde, 2017; Table 2). Specifically, undamaged, fully green floating leaves randomly sampled outside the plots were taken to the laboratory where both length and area were measured to establish relationships of the form: (1) AL=ciL2

Table 2 Length-area regression equations for the leaves of the three study species.

Length-area regression equations of undamaged fresh green leaves of the three study species.

Species	N	Regression equation	r2	p	
Nuphar lutea	37	A = 0.623L2	0.99	<0.001	
Nymphaea alba	84	A = 0.788L2	0.98	<0.001	
Nymphaea candida	10	A = 0.695L2	0.99	<0.001	
Notes.

N number of leaves used to determine the equation coefficients

A leaf area

L leaf length

where:

A(L) = potential leaf area at length L (cm2)

L = leaf length from the leaf tip to a basal lobe tip (cm)

ci = regression coefficient of species i

i = species (Nuphar lutea, Nymphaea alba, Nymphaea candida)

Study design and data collection

Six representative plots of 1 m2 were laid out in the center of mono-specific stands, each containing one rhizome apex per plot. A non-destructive method was used to tag all floating leaves individually within the plots (Klok & Van der Velde, 2017). Newly unrolled leaves were tagged with uniquely numbered Rotex tape fixed around the petiole just under the leaf blade. This enabled us to collect data during the full life-span of the leaves. Each plot was bordered by a square perforated PVC tube frame, held approximately 15 cm below the water surface by cork floaters and anchored to the bottom by four bricks. This set-up does not affect the unrolling of floating leaves in the plots. All leaves having their petioles within the frame were counted and measured. A leaf was considered present as long as, after partial degradation and disintegration, tissue of the lamina was connected to the petiole in the case of OW and HW. In VG a leaf was considered ‘gone’ when it was completely brown, dead and submerged, or when it had disappeared.

All leaves within the plots were inspected and measured at weekly intervals during the growing season, typically from April until November. Site visits involved tagging newly unrolled leaves, counting the number of leaves, measuring leaf length from the leaf tip to one of the basal lobe tips and visually estimating different types of initial decomposition expressed as percentage of the potential leaf area of each leaf. Leaves showing several types of damage were harvested outside the plots to be photographed in the laboratory.

Results

Leaves developed during 53 to 73% of the vegetation period of 135 to 199 days (Klok & Van der Velde, 2017). Loss of leaf tissue tended to increase during the vegetation period (Fig. 2; Table 3). In the hardwater lakes (OW and HW), leaf area loss by damage of Nuphar lutea and Nymphaea alba was less than 20% of the total potential leaf area until mid-September, but increased to more than 50% thereafter. Leaf area loss by damage of Nymphaea candida (HW) was less than 10% of the potential area in the beginning and increased to almost 20% in September–October. In the acidic moorland pond (VG) leaf area loss was minimal as these leaves did not disintegrate.

Figure 2 Photosynthetic leaf area loss by external causes in time per plot.

Seasonal changes in the loss of photosynthetic leaf area caused by external factors, shown as the difference between the actual area (lower line) and the potential area (upper line) in six field plots. (A) Nuphar lutea, HW, 1977, (B) Nuphar lutea, OW, 1977, (C) Nymphaea alba, OW, 1977, (D) Nymphaea candida, HW, 1977, (E) Nuphar lutea, VG, 1988, (F) Nymphaea alba, VG, 1988.

Table 3 Summary characteristics of waterlily stands.

Summary characteristics of waterlily stands in three water bodies of The Netherlands.

Site	Species	Year	Vegetation period	Growth period	Total number of leaves (m−2)	Total potential leaf area (cm2)	
			Time span	Days	Time span	Days			
HW	Nuphar lutea	1977	May 10–Nov 24	199	May 10–Sep 13	127	77	49,674	
OW	Nuphar lutea	1977	May 11–Nov 1	175	May 11–Sep 7	120	59	39,898	
VG	Nuphar lutea	1988	Apr 28–Oct 27	183	Apr 28–Sep 8	134	22	8,440	
HW	Nymphaea candida	1977	Jun 7–Oct 19	135	Jun 7–Aug 16	71	43	11,185	
OW	Nymphaea alba	1977	May 11–Nov 6	180	May 11–Sep 7	120	108	53,035	
VG	Nymphaea alba	1988	Apr 28–Oct 27	183	Apr 28–Sep 8	134	80	23,053	
Notes.

HW Haarsteegse Wiel

OW Oude Waal

VG Voorste Goorven

Causes and impacts of initial decomposition

The causes of damage classified in the present study are senescence, frost, hailstones, dehydration, mechanical damage, bird scratches, feeding waterfowl (Fulica atra L. and Gallinula chloropus L., Rallidae), pond snails (Lymnaea sp., Lymnaeidae, Gastropoda), water-lily reed beetle (D. crassipes F., Chrysomelidae, Coleoptera), adults and larvae of the water-lily leaf beetle (Galerucella nymphaeae L., Chrysomelidae, Coleoptera), a weevil (Bagous rotundicollis Bohemann, Curculionidae, Coleoptera), larvae of the aquatic moth brown china mark (Elophila nymphaeata (L.), Crambidae, Lepidoptera), larvae of a dung fly (Hydromyza livens (Fabricius), Scathophagidae, Diptera), chironomid larvae (Chironomidae, Diptera), including Endochironomus spp. and T. intextus (Walker), a phytopathogenic fungus (Colletotrichum nymphaeae (Pass.) Aa) and an oomycete (Pythium sp.), and finally microbial decay (Fig. 3, Table 4). In some cases, specific causes could not be identified.

Figure 3 Relative contributions to leaf damage by external causes per plot in time.

Seasonal changes in the relative contributions to leaf damage by external causes in six field plots. (A) Nuphar lutea, HW, 1977, (B) Nuphar lutea, OW, 1977, (C) Nymphaea alba, OW, 1977, (D) Nymphaea candida, HW, 1977, (E) Nuphar lutea, VG, 1988, (F) Nymphaea alba, VG, 1988.1 = Frost, 2 = Dehydration, 3 = Mechanical damage, 4 = Scratches, 5 = Damage by Elophila nymphaeata, 6 = Consumption by Fulica atra, 7 = Consumption by snails, 8 = Consumption by Donacia crassipes, 9 = Consumption by Bagous rotundicollis, 10 = Consumption by Galerucella nymphaeae, 11 = Mining by Hydromyza livens, 12 = Mining by Chironomidae, 13 = Mining by Endochironomus, 14 = Fungi, 15 = Microbial decay, 16 = Unknown causes.

Table 4 Damage to leaves during initial decomposition.

Prevalence of different causes of leaf damage during initial decomposition of floating leaves at six plots in three water bodies located in The Netherlands. The total number of leaves and the total potential area of leaves per plot are listed in Table 3.

Cause of damage	Percentage of leaves affected	Percentage of potential area affected	Photosynthetic area lost (cm2)	
	(1)	(2)	(3)	(4)	(5)	(6)	( 1)	( 2)	( 3)	( 4)	( 5)	( 6)	(1)	(2)	(3)	(4)	(5)	(6)	
							av.	max.	av.	max.	av.	max.	av.	max.	av.	max.	av.	max.							
Senescence	79	92	91	84	78	64	6.3	40.0	6.2	19.0	4.8	23.5	10.9	39.0	5.4	35.0	2.9	15.7	4278	2508	1868	2181	4727	2748	
Frost	–	2	–	–	–	–	–	–	<0.1	0.8	–	–	–	–	–	–	–	–	–	5	–	–	–	–	
Hail stones	–	–	–	–	–	–	–	–	–	–	–	–	–	–	–	–	–	–	–	–	–	–	–	–	
Dehydration	23	37	–	9	28	6	0.5	5.0	1.0	6.9	–	–	0.1	0.6	0.6	7.8	0.2	8.0	384	603	–	9	854	48	
Mechanical damage	78	47	–	74	80	–	1.1	8.8	1.2	10.0	–	–	0.8	3.3	1.5	10.9	–	–	546	577	–	95	1118	–	
Bird scratches	83	59	–	84	77	–	0.7	1.0	0.5	1.0	–	–	0.6	1.0	0.6	1.0	–	–	382	223	–	83	386	–	
Consumption by coots	36	14	–	12	50	–	0.8	10.0	0.6	18.0	–	–	0.1	0.9	0.6	3.0	–	–	385	204	–	14	442	–	
Consumption by pond snails	56	12	–	12	13	–	2.5	10.0	0.4	5.4	–	–	0.3	5.0	0.3	8.0	–	–	1113	203	–	26	120	–	
Consumption by reed beetles	65	63	73	70	54	–	0.6	2.0	0.6	1.8	0.9	2.0	0.6	1.2	0.4	1.6	–	–	375	285	64	74	324	–	
Consumption by waterlily beetles	–	–	–	–	–	24	–	–	–	–	–	–	–	–	–	–	0.3	2.7	–	–	–	–	–	85	
Consumption by weevils	–	–	–	–	–	29	–	–	–	–	–	–	–	–	–	–	0.2	1.0	–	–	–	–	–	63	
Consumption and damage by the brown china mark	10	3	–	–	6	–	0.4	5.0	0.1	3.6	–	–	–	–	0.1	3.9	–	–	144	43	–	–	66	–	
Mining by a dung fly	65	69	73	–	–	–	1.3	6.5	1.1	4.0	1.3	3.5	–	–	–	–	–	–	786	516	119	–	–	–	
Mining by chironomids	14	2	–	2	6	–	0.2	5.0	<0.1	1.0	–	–	<0.1	0.4	<0.1	1.0	–	–	99	7	–	3	33	–	
Mining by Endochironomus	5	–	50	12	25	23	<0.1	1.2	–	–	1.1	5.0	0.1	1.0	0.3	1.8	0.5	5.4	34	–	99	13	181	110	
Infection by Pythium “type F”	86	92	77	–	–	–	4.2	11.8	6.1	12.9	1.0	4.9	–	–	–	–	–	–	2879	3153	277	–	–	–	
Infection by Colletotrichum nymphaeae	–	–	–	79	53	94	–	–	–	–	–	–	6.7	17.9	6.1	21.7	2.1	8.8	–	–	–	3274	11464	767	
Microbial decay	56	86	59	56	72	60	4.9	26.3	9.7	26.1	4.6	80.3	0.4	5.3	2.8	26.8	1.3	64.3	8803	11844	766	182	5634	6314	
Unknown causes	65	5	–	19	34	–	7.2	33.3	0.1	1.0	–	–	1.0	26.7	1.6	40.0	–	–	3888	20	–	115	1235	–	
Notes.

av. average

max. maximum

1 Nuphar lutea, Haarsteegse Wiel, 1977

2 Nuphar lutea, Oude Waal, 1977

3 Nuphar lutea, Voorste Goorven, 1988

4 Nymphaea candida, Haarsteegse Wiel, 1977

5 Nymphaea alba, Oude Waal, 1977

6 Nymphaea alba, Voorste Goorven, 1988

Senescence

Senescence is visible by the change in leaf colour from green to yellow, indicating that chlorophyll is degraded. The extent of yellow areas reached its maximum towards the end of the growing season. In October the percentage of affected leaves was 100%; however, the yellow surface area was generally around 10% and loss of green photosynthetic leaf tissue ranged between 10 and 20% of the total leaf loss. The extent of leaf area turned yellow decreased over time, since brown leaf areas leading to microbial decay became increasingly dominant (Fig. 4; Table 4).

Frost

Frost in early spring can damage the tips of young leaves sticking out of the water. As a result, such leaves can lose up to one third of their area (Fig. 5). However, the effect on the total leaf surface area was less than 5%.

Figure 4 Senescence and microbial decay.

(A, B, C) show the sequence of colour changes from green living tissue (dark grey areas) to senescent tissue (light grey areas) and areas of microbial decay (black areas) on Nymphaea candida, photographed with translucent light.

Figure 5 Frost.

Symptoms of frost damage of a whole N. lutea leaf (A) and the tip of a Nuphar lutea leaf sticking out of the water which could be damaged by frost (B).

Figure 6 Hailstones.

(A, B) show symptoms of damage by hailstones (white arrows) and snails (black arrows) on Nymphaea alba.

Hailstones

Occasional hailstone showers damage the floating leaves by penetrating the leaf and leaving typical Y-shaped scars (Fig. 6). Leaf area damaged by hail was minimal.

Dehydration

High winds often lift floating leaves above the water surface and may flip them over. Subsequently, those leaves are exposed to air, particularly the leaf margins, leading to leaf desiccation (Fig. 7). The effect of desiccation stress on leaf surface area was generally less than 5%.

Mechanical damage

This type of damage is caused by wind and wave action resulting in cracks in the leaf tissue or lost leaves when the petiole breaks (Fig. 7). Lost leaves were ascribed to unknown causes. For Nuphar lutea at HW, Nymphaea alba at OW and Nymphaea candida at HW, the percentage of leaves affected over the whole vegetation period ranged from 60–80%. Nuphar lutea at OW showed peaks of 90% in spring, 70% in autumn and 10% in summer. In contrast, Nuphar lutea at VG and Nymphaea alba at VG showed no mechanical damage.

Bird scratches

Scratches are often caused by the claws of birds, mostly coots (F. atra) but also the common moorhen (G. chloropus), as they walk or run over the leaves (Fig. 8). In general, the scratches are straight and affect only the epidermis of the leaf, but angle-shaped cuts due to claws penetrating the leaf tissue also occur. The affected leaf surface area was low, generally below 5%, although a high percentage of leaves was affected, sometimes up to 100% for all plots at HW and OW. In contrast, the plots at VG showed no scratches.

Figure 7 Wind and wave action.

Uplifted leaves as a result of wind and wave action (A, B), leading to mechanical damage as well as to dehydration by air and sun exposure, as shown at the leaf margin of Nuphar lutea (C).

Figure 8 Bird scratches.

Bird scratches caused by the claws of F. atra or G. chloropus, damage caused by Pythium “type F”, and dehydration of the leaf margin.

Consumption by coots

Consumption of leaf tissue by coots can be recognized by missing parts in the form of triangular areas at the edge of leaves. Sometimes major parts of leaves are consumed. Generally, prints of the beak are visible around the consumed areas (Fig. 9). Nevertheless, the overall effect on total leaf surface area was minimal. The plots at VG showed no damage by coot consumption.

Consumption by pond snails

A major cause of damage on fresh leaf tissue is caused mainly by Lymnaea stagnalis L., to a lesser extent also by other lymnaeids. Pond snails consume folded leaves still under water. Rows of holes can then be seen in the unrolled leaf blades, large near the edge and smaller towards the center of the leaf (Fig. 10). Lymnaeid and other freshwater pulmonate snails show a preference for decaying leaf material, such as areas infected by fungi. Damage by snails was generally an important cause of damage during the whole period for both Nuphar lutea and Nymphaea alba, contributing up to 20% to the total leaf area loss in HW.

Figure 9 Consumption by coots.

(A, B, C) show damage and leaf area loss due to the consumption of tissue by Fulica atra on Nymphaea alba.

Figure 10 Consumption by pond snails.

The pond snail L. stagnalis (A) causing leaf damage of Nymphaea alba, which is visible as rows of holes caused by the snail before leaf blades unroll (B, C).

Consumption by water-lily reed beetles

Both Nuphar and Nymphaea spp. are host plants of the water-lily reed beetle D. crassipes. The adult beetles live on the upper side of floating leaves where they feed on leaf tissue (upper epidermis, parenchyma and lower epidermis). The leaf areas removed as a result of tissue consumption by these beetles are round to oval. Eggs are deposited in two or three rows on the leaf underside. To this end, the beetle gnaws a round or oval hole in the leaf, then sticks its abdomen through the hole to reach the leaf underside and oviposit (Fig. 11). The percentage of leaf area damaged by reed beetles was minimal.

Figure 11 Consumption by the water-lily reed beetle D. crassipes.

Floating-leaf consumption by the water-lily reed beetle D. crassipes. (A) Size of consumed spots and of egg deposition holes made by imagines of the beetle on leaves of Nuphar lutea, (B) eggs deposited at the underside of a Nuphar lutea leaf, (C) imago on Nymphaea alba, (D, E, F, G) leaves of Nuphar lutea damaged as a result of consumption by D. crassipes.

Consumption by water-lily leaf beetles

The water-lily leaf beetle (G. nymphaeae) completes its full life cycle on the upper surface of floating leaves. Both adult beetles and larvae feed on the upper epidermis and palisade and sponge parenchyma. The larvae create irregular trenches on the surface, leaving the lower epidermis of the leaf intact and depositing their faeces in the trenches. The resulting pattern of leaf tissue damage is easily recognized. The adult beetles consume smaller areas (Fig. 12). Damage was only found in N. alba at VG, where leaves started to be affected in mid-June, rising to 30–40% between August and October and reaching a sharp peak of 60% in mid-October. The percentage of lost leaf area was minimal.

Figure 12 Consumption by the water-lily leaf beetle G. nymphaeae.

Consumption of floating leaves by the water-lily leaf beetle G. nymphaeae. (A) Eggs, (B) larvae and pupae, (C) imago with consumption spots, (D) typical damage patterns caused by larvae on Nymphaea alba, and (E, F) damage patterns caused by larvae and imagines on Nuphar lutea.

Consumption by weevils

The adults of B. rotundicollis scrape off areas of leaf tissue (ca. 1 cm diameter) from the underside of floating leaves near the margin. Only the lower epidermis and sponge parenchyma are consumed, whereas the palisade parenchyma and upper epidermis remain intact (Fig. 13). Damage by weevils was found only in N. alba at VG, with up to 30% of these leaves being affected. Leaf area loss was minimal.

Figure 13 Consumption by the weevilB. rotundicollis.

Consumption by the weevilB. rotundicollis. (A, B) imago and (C) damaged spots indicated by white arrows along the margin on the underside of a leaf.

Consumption and damage by the brown china mark

The caterpillar of the aquatic moth E. nymphaeata damages floating leaves in two ways, by leaf tissue consumption and by cutting out oval leaf patches that they can attach to the underside of a floating leaf to make a shelter. They can also spin two patches together to construct a floating shelter (Fig. 14). The effect of these activities on leaf surface area was low, at most 5%. Nymphaea candida at HW, Nuphar lutea at VG and Nymphaea alba at VG were not damaged by the moth.

Figure 14 Consumption and damage by the brown china mark E. nymphaeata.

Consumption and damage by caterpillars of the brown china mark E. nymphaeata on Nymphaea alba. (A, B) Caterpillar in a free-floating shelter composed of two pieces of floating leaf, (C) adult moth on a leaf, (D, E) damage on floating leaves of Nymphaea alba.

Mining by a dung fly

Larvae of the dung fly Hydromyza livens only occurred in Nuphar leaves, where they mine and consume leaf tissue. Eggs are laid at the underside of the leaves. For that purpose the fly goes underwater, following the dichotomous veins on the underside of the leaves till it reaches the midrib to lay an egg. The newly hatched larvae immediately start to mine the leaf tissue. The mine track has a characteristic shape as the larvae first move from the midrib towards the margins of the leaf, then turn to continue mining in parallel to the leaf margin, then turn again towards the midrib and mine further into the petiole where they pupate. This creates a breaking point where the leaf blade can detach and float away (Fig. 15). Overall, the effect of dung flies mining the leaves was less than 8%.

Figure 15 Mining by larvae of the dung fly Hydromyza livens.

Mining by larvae of the dung fly Hydromyza livens. (A) Eggs of Hydromyza livens on the underside of a Nuphar lutea leaf, (B) scanning electron micrograph of the head of a larva, (C) drawing of an imago, (D, E) larval mine tracks on Nuphar lutea and infection by Pythium spec. (scattered small spots). Photos of leaves taken with translucent light.

Mining by chironomids

Larvae of some Chironomidae mine their way through the leaf tissue by consuming particular tissue layers while leaving the upper and lower epidermis unaffected for protection. Typical damage on Nuphar leaves is caused by larvae of T. intextus. These larvae mine leaves still folded underwater, resulting in rows of small holes that become visible when the floating leaves unroll at the water surface (Fig. 16). Also observed at the study sites were larvae of C. trifasciatus (Meigen) (Fig. 17), which makes an open mine by removing the upper epidermis while leaving the lower epidermis intact. The species was observed in OW to cause some damage at the leaf margins of Nuphar lutea in the neighbourhood of Nymphoides peltata (Gmel.) O. Kuntze, its main food plant. Overall, however, the impact of these chironomid species on floating leaves was minimal.

Figure 16 Mining by larvae of the chironomid T. intextus.

(A, B) show damage by mining by larvae of the chironomid T. intextus on Nuphar lutea.

Figure 17 Typical mining patterns by larvae of C. trifasciatus.

Typical mining patterns caused by larvae of C. trifasciatus (Chironomidae) on floating leaves. Patterns in the centre of a leaf blade (A) and near the leaf margin (B, C).

Mining by Endochironomus spp

Larvae of these midges mine in floating leaves. The mines could clearly be distinguished from those of other Chironomidae described above, since they appear on the upper side of the floating leaves as straight dark stripes (Fig. 18). The total effect on the decomposition of floating leaves was minimal.

Infection by phytopathogens

The leaves of Nuphar lutea were infected by the oomycete Pythium “type F” (Fig. 19) and the leaves of Nymphaea alba and Nymphaea candida by the fungus Colletotrichum nymphaeae, the causative agent of leaf spot disease (Fig. 20). The percentage of damaged surface area was about 15% for Pythium and up to 55% for Colletotrichum.

Microbial decay

The resistance of a leaf to microbial infection quickly disappears during senescence, facilitating microbial decay (Fig. 4) as indicated by a change in leaf colour from yellow to brown and the softening of leaf tissue by maceration. The affected surface area rose to 15–25%, with an exceptional extent of 60% reached in Nymphaea candida at HW at the very end of the growing season.

Figure 18 Typical mining patterns by larvae of Endochironomus spec.

Typical mining patterns by larvae of Endochironomus spec. (Chironomidae). Patterns on a leaf (A, B) and near the leaf margin (C).

Figure 19 Damage by Pythium “type F”.

Damage by Pythium “type F” on Nuphar lutea (A-H), photographed with translucent light.

Figure 20 Damage by Colletotrichum nymphaeae.

Damage caused by Colletotrichum nymphaeae on Nymphaea alba (A, B, C, D) and infected spots consumed by snails (A).

Unknown causes

Missing leaves or parts thereof can result from various types of damage, including animal consumption and mechanical damage. Missing leaf material where the cause of loss could not be determined was registered under unknown causes. These causes include leaves disconnected from their petioles and scattered by wind and wave action, occasionally accounting for up to 60% of lost area for Nuphar lutea at HW, Nymphaea alba at OW and Nymphaea candida at HW. However, such losses were rare for Nuphar lutea at OW and VG and Nymphaea alba at VG.

Discussion

Senescence

Newly unrolled leaf blades of waterlilies are fully green and hydrophobic due to a thick epicuticular wax layer. This waxy layer gradually erodes during senescence and as cellulolytic and other bacteria and fungi colonize the leaf tissue (Howard-Williams, Davies & Cross, 1978; Robb et al., 1979; Rogers & Breen, 1981; Barnabas, 1992). Senescence starts shortly after the first leaves are fully grown and continues throughout the growth period. During senescence, an orderly physiological process controlled by the plant itself, the leaves turn from green to yellow, and ultimately to brown. Concomitant microbial decay softens the leaves.

Infection by phytopathogens and microbial decay

In Nuphar both microbial decay and infection by the phytopathogenic oomycete Pythium sp. “type F” were important from the start of the season. In Nymphaea, infection by the phytopathogenic fungus Colletotrichum nymphaeae also started early and increased in importance towards the end of the season. In general, microbial decay and phytopathogenic infection gradually increased in importance, whereas most other causes of damage diminished over time.

Weather conditions

Minor causes of leaf impairment occurring once during spring were frost damage of the first newly unrolled leaves and hailstones. Hailstones hardly caused leaf area loss. High solar radiation and air temperature dehydrated leaves that had been flipped over, with the impact being high in HW and OW but not in the wind-sheltered VG. Prolonged cloudy and wet weather imposes stress on waterlilies by weakening the defense of leaves due to reduced solar radiation, and thus promoting heavy infection and damage by phytopathogens (Van der Aa, 1978). One mechanism is that poor light conditions reduce the content of phenolics with fungistatic properties in the leaf tissue (Vergeer & Van der Velde, 1997), which turns mature leaves vulnerable to infection.

Damage by animals

Causes of damage by insects were similar for Nymphaea and Nuphar with the exception of Hydromyza livens and T. intextus, which appear to be specific for Nuphar (Brock & Van der Velde, 1983; Van der Velde & Hiddink, 1987). Some species such as B. rotundicollis (Van der Velde, Kok & Van Vorstenbosch, 1989) and D. crassipes (Gaevskaya, 1969) exclusively feed on Nymphaeaceae. Other species such as Galerucella nymphaeae and Elophila nymphaeae feed on both floating-leaved and emergent macrophytes (Gaevskaya, 1969; Lammens & Van der Velde, 1978; Pappers et al., 2001). C. trifasciatus primarily causes damage on leaves of Nymphoides peltata (Lammens & Van der Velde, 1978) but was also observed to damage nearby Nuphar lutea leaves (Van der Velde & Hiddink, 1987).

In VG, damage was mainly caused by phytophagous insects consuming floating leaf tissue, particularly herbivorous beetles, fly larvae and mining chironomid larvae. Leaf disintegration was hardly observed in the acidic VG, which was also the site most sheltered against wind and wave action by a surrounding forest. Protection from wind and wave action allowed the water-lily leaf beetle Galerucella nymphaeae to cause extensive damage, because the wind blows them from the leaves and they float away as a result of wave action. Although this species spares the lower epidermis of their tracks, the epidermis becomes vulnerable to microbial attack and thus disappears at a later stage (Wesenberg-Lund, 1943; Roweck, 1988). As observed in the present study, the minor leaf area loss by the beetle and its larvae is succeeded by damage caused by fungi, oomycetes or bacteria (Wallace & O’Hop, 1985). The damaged areas characterized by regular margins made by adult Galerucella nymphaeae are distinct from those made by adult D. crassipes where the margins of damaged areas are rather irregular (Roweck, 1988). Galerucella nymphaeae was absent in the two water bodies frequently exposed to strong wind.

Consumption by snails was restricted to the two hardwater lakes, since they require calcium to build their shells. Snails at those sites prefer consuming microbially colonized, decaying parts of the leaves (Kok, 1993).

Nymphaea candida (HW) showed an increase in scratches by bird claws towards the end of June, which may have been due to young coots. High densities of waterfowl at HW and OW are facilitated by the surrounding meadows where birds graze during winter.

pH and alkalinity

Decomposition of leaves was slowed down at the acidic site (VG). Such water bodies are characterized by a very low alkalinity and high Al concentrations of the water, as well as low pH (Leuven, Van der Velde & Kersten, 1992). A laboratory study in chemostats with synthetic media showed that pH, Al and HCO3− concentrations clearly influence the decomposition and chemical composition of leaf blades of floating-leaf plants, with low pH and elevated Al concentrations inhibiting and high bicarbonate concentrations (alkalinity) stimulating decomposition (Kok, Meesters & Kempers, 1990). Al is toxic to microorganisms and low pH slows down leaf disintegration by inhibiting cell-wall degradation by microbial pectin-degrading exoenzymes and xylanase (Kok & Van der Velde, 1991). At low pH, tannins accumulate in the slowly decomposing leaf material, microbial colonization is inhibited and maceration of the leaf tissue is reduced, resulting in a low-quality food resource for detritivores (Kok et al., 1992). The occurrence of detritivores is also inhibited by high Al concentrations and low pH (Kok & Van der Velde, 1994). Finally, fungal degradation of major groups of structural carbohydrates is inhibited by low pH (Kok, Haverkamp & Van der Aa, 1992).

Harvested fresh and decaying leaf blades of Nymphaea alba placed in litter bags in the field and in the laboratory showed lower leaf area loss under acidic conditions in a moorland pond (VG) than in a eutrophic, hardwater oxbow lake (OW), and results under laboratory conditions mimicking differences in water chemistry were similar (Brock, Boon & Paffen, 1985). Depending on water chemistry, mass loss was pronounced and organic matter chemical composition changed rapidly during the first 10–30 days, followed by an accumulation of structural plant polymers such as cellulose, hemicellulose and lignin. The disappearance of those fractions was dependent on the water quality of the water body (Brock, Boon & Paffen, 1985).

In conclusion, the present study shows that the decomposition pattern of Nuphar lutea was similar in the two hardwater lakes, and differed from those of Nymphaea alba and N. candida. In the acidic VG, the effect of leaf damage on leaf area loss was minimal for both Nuphar lutea and Nymphaea alba.

Supplemental Information

Supplemental Information 1 Raw data on initial decomposition of floating leaves

Click here for additional data file.

We thank M Ankersmid, R Kwak, R de Mooij, H Peeters, F Verhoeven, V Vintges and CJ Kok for collecting field data, RPWM Jacobs and HA van der Aa for identifying oomycetes and fungi, respectively, W Lemmens for help with data analysis, WJ Metzger for English language corrections, and the editor Mark Gessner, reviewer Manuela Abelho and one anonymous reviewer for constructive comments that very much improved the manuscript.

Additional Information and Declarations

Competing Interests

Author Contributions

Data Availability

The authors declare there are no competing interests.

Peter F. Klok analyzed the data, contributed reagents/materials/analysis tools, prepared figures and/or tables, authored or reviewed drafts of the paper, approved the final draft.

Gerard van der Velde conceived and designed the experiments, performed the experiments, contributed reagents/materials/analysis tools, authored or reviewed drafts of the paper, approved the final draft.

The following information was supplied regarding data availability:

Raw measurements are available in the Supplemental Files.

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
