# Peer review of "Initial decomposition of floating leaf blades of waterlilies: causes, damage types and impacts"

_PeerJ, doi:10.7717/peerj.7158_

## Round 0.1 · original submission · Major Revisions

Thank you for submitting your synthesis paper on many years of investigations into the initial decomposition of water lilies. As you will see, both reviewers see merit in your paper but are very critical about its presentation. Consequently, I would like to invite you to revise the paper very thoroughly to improve organization of the content, the flow of thoughts, and the wording. This should result in a significantly tightened manuscript. The many detailed comments of the reviewers will be an excellent guide. I look forward to receiving a revised version of the manuscript in due course. The level of required changes is such that I will then have to send the paper for review again.

·

Basic reporting

1.The language can be improved, as suggested in several notes on the attached manuscript. Many explanations through the text are given in parentheses. The manuscript would benefit with the removal of the parentheses; e.g. line 57: «…of agriculture, horticulture and forestry (for which phytopathology is a main discipline),…» replaced by «…of agriculture, horticulture and forestry, for which phytopathology is a main discipline,…»
2.The introduction has enough background to show the context of the study. However, the contents of the sub-section «Initial decomposition and causes» in Materials and Methods also belong to Introduction, after the explanation of the phases of decomposition.
3.The literature is well referenced and relevant. However, most of the references are from the 1070s-1990s, maybe due to the fact that the data was collected in 1977 and 1988. Also, there are too many references for the same subject in some parts of the text as in lines 109-112.
4.The structure of the manuscript conforms to PeerJ standards, but the contents of some sections must be reorganized. The sub-section «Initial decomposition and causes» in Materials and Methods belong to Introduction. While reporting the results, the authors discuss the results and cite other authors and during the Discussion, only one author is cited. In the conclusions, the authors continue the discussion. To conform to the PeersJ standars, all discussion should be allocated to the Discussion section, the contents of lines 378-383 (Discussion) are an introductory synthesis of the results and should be in fact the beginning of the results section. In order to fully understand that the work was carried out during 1977 and 1988, this should be referenced in the beginning of the Materials and Methods, before using, for instance, (HW, 1977).
5.Figures and tables: table 1 must be improved in the information provided, namely: is depth an average? If so, a range or at least maximum depth should also be provided. Still in table 1, the values of the physical characteristics are not uniform: there is no range for the first two data points, and the number of measurement should also be provided Table 2 is not necessary since its contents are already in text. Figure 4 is not necessary since it does not illustrate any result. Legend of figure 14 does not match its contents and figure 14 not necessary in the manuscript. Regarding the rest of the figures, they are mostly photographs and drawings which, in some cases, would benefit from better quality if they are to be used as a guide by other authors.
6.Raw data is supplied as an Excel file, it opens, but as far as I understand, only data for one of the species in one of the years is provided. Also, it is not clear what the numbers 3-21 mean

Experimental design

No comment

Validity of the findings

No comment

Additional comments

This is a descriptive manuscript reporting the causes and effects of initial leaf decomposition in floating leaves of 3 species in 3 lakes. Data is old and was collected in different occasions (with a difference of eleven years) in the 3 lakes, and the sampled species are not the same in the 3 lakes. The lakes differ in their abiotic variables, sampling dates are not uniform and there is no true replication, and it is clear that the manuscript reports the results of two different studies, one performed in 1977 and the other in 1988. However, given the descriptive characteristics of the study and given that the study does not compare either the lakes or the plant species –although in the discussion, you address the effects of some environmental variables on the results, the timing of sampling and the sampled species are not truly important. In fact, other investigators may use the descriptive nature of the results of the study as a kind of key for identification of the causes of initial decomposition in floating leaves. However, to fulfil this objective the manuscript has to be greatly improved.

Reviewer 2 ·

Basic reporting

See General comments to authors below.

Experimental design

The study is original and within the aims and scope of the journal. Overall experimental design seems sound and will fill a gap in knowledge concerning floating leaf macrophyte damage and decomposition. However, as stated in the general comments to authors below, I feel the manuscript is in need of editing to provide greater clarity

Validity of the findings

See General comments to authors below.

Additional comments

“Initial decomposition of floating leaf blades of waterlilies: causes, damage types and their impact” by Klok and van der Velde.

This manuscript presents an overview of a series of field studies that investigated the biotic and abiotic factors involved in the initial decomposition of floating leaf blades of Nuphar lutea, Nymphaea alba, and Nymphaea candida. Here, floating leaf blades of N. lutea, N. alba, and N. candida were examined at three freshwater sites that differed in environmental conditions (trophic status, pH and alkalinity. Floating leaf blades were tagged/numbered within replicate plots and leaf length, percentages and types of damage to, and decay of, each leaf was measured and estimated weekly for all plots during the growing season. The manuscripts proceeds to describe the initial decomposition stages of leaves and the various causes (biotic and abiotic) that are chiefly responsible.

Overall Comment
Overall, I found that this manuscript is interesting and contains meticulous data on the types of damage to, and decay of, the leaf blades of water lilies. However, I must admit from the outset that there is a lot going on in the manuscript that makes any full evaluation quite difficult. In my opinion, this manuscript is not yet ready for publication in its present form. I believe the underlying premise of the study is sound and interesting, and will add to our understanding of decay patterns (and causes) in water lilies. My main concern with this paper is with the overall writing and logical flow of the manuscript. In my opinion, this manuscript is just not ready for publication and will require a very careful, lengthy revision. A large part of the writing within the manuscript is in need of editing. There are several confusing words, sentences and passages throughout the manuscript that are not well written or connected. As a consequence, the manuscript lacks clarity and overall flow of thought in many areas. Hence important information and its significance are lost or obscured. I would have so many specific comments concerning the text that it would be just pointless to list them all.

I will give just a few examples in my specific comments below:

Abstract – In my opinion the abstract needs to be restructured. I am ok with the first paragraph, with some editing se below. However, the second paragraph needs to be entirely rewritten for the following reason:

The authors state - “Initial decomposition and its various causes are depicted and described. Also leaf damage with respect to the potential leaf area per species per plot, contributions to leaf damage by external causes, leaf loss in time and succession of damage causes are presented.”
Please do not tell me in the abstract of the manuscript that data will be depicted, described or presented within the manuscript. Please depict, describe or present a synopsis of the actual data in the manuscript within the abstract.

Abstract – first paragraph. How about - Initial decomposition (i.e. leaf damage and loss) of large floating leaved macrophytes, such as waterlilies, can be studied well: the turnover of floating leaf blades is low and leaves can persist for a relatively long time. In the present study, the initial decomposition patterns of floating leaf blades of Nuphar lutea (L.) Sm., Nymphaea alba L. and Nymphaea candida Presl, were examined at three freshwater sites differing in water quality, such as nutrient status, pH and alkalinity. Floating leaf blades of each species were tagged and numbered within established replicate plots and the leaf length, percentages and types of damage to and decay of leaves measured and estimated weekly throughout the growing season.

Abstract – second paragraph. Here is an example of what I mean when a passage is not very well connected in terms of flow of thought. Below I list the sentences in the second paragraph in order. Each addresses a quite separate topic, giving the reader an impression that the paragraph just jumps around and is a random collection of statements. It would be much better to restructure this paragraph for clarity (particularly the 4th sentence which is quite long). Also, the authors state (4th sentence) “…followed by cellulolytic bacteria, or fungi…followed by microbial decay”. Cellulolytic bacteria or fungal decay is microbial decay, so I do not have any idea what the authors are trying to convey here.

“Only a few damage causes had a significant impact on leaf damage and leaf loss: autolysis, fungi, snails and mechanical damage.

The floating leaves offer food for a series of specialized insects consuming leaf area from below the water surface, from the upper surface or by mining the leaf tissue.

Waterfowl (e.g. Rallidae) consume leaf parts and walk on the leaves scratching the upper surface.

Several forms of succession of damage can be distinguished such as erosion of the wax layer, followed by cellulolytic bacteria, or fungi, followed by snails, or mechanically damaged leaves (by wind and wave action, desiccation and hail stones), followed by biotic causes and decay, or autolysis, followed by microbial decay, followed by tissue removal by snails, followed by breaking up of leaves.

In alkaline waters the seasonal patterns of initial decomposition differed between Nymphaea and Nuphar.”

Introduction- first paragraph

“Already during their development plant leaves are exposed to abiotic factors (such as weather conditions, causing physical damage, fragmentation and drying out) as well as biotic factors (such as infection by fungi and viruses, herbivores, and animals using these parts of the plant in various ways). This exposure is well-known for crops and ornamental plants as it causes economic damage. Plant resistance depends on age and plant injuries (Kennedy & Barbour, 1992). Initial causes of decomposition (when the leaves are still connected to the plant) precede the process of leaf material entering the decomposition cycle on the soil. In ecological studies much attention is paid to the latter because these soil processes are important for the biogeochemical cycles. However, with the exception of agriculture, horticulture and forestry (for which phytopathology is a main discipline), much less attention is paid to the first process, in particular for aquatic macrophytes.”

I find the sentences in this paragraph (particularly the first three) to be a rather awkward transition between each other. I am just what the authors want to convey here? The first sentence sets the idea of leaf damage in terms of biotic and abiotic factors (which I get), but following sentences confuse me and don’t seem to fit the idea being proposed, particularly “Plant resistance depends on age and plant injuries (Kennedy & Barbour, 1992).” This sentence is just kind-of stuck in there with no logical transition or flow of thought.

These are just two examples, but there are several more scattered throughout the manuscript.

There are also some usage of words that seem odd to me…like “vital”, “sapropelium”, and “ “percentual”. Why not just use living, organic matter (or detritus) and percent (or percentage). These words are better suited for the manuscript as they are more widely accepted, in my opinion.

Results – Causes of initial decomposition and their impact.

In my opinion, a lot of this material could be consolidated and summarized better. Perhaphs having a section labeled biotic and abiotic, where the influence of such factors are described. Right now, I feel it is too long and is hard to get through and pick out the important information. This section should be restructured.

Discussion and Conclusion – In comparison to other sections in the manuscript. I feel that this section is under-developed. In addition, some of this information in the results could be moved here and put in to the context of other studies.


Figures and Tables – 23 Figures and 5 Tables. Seems like a lot, but in general I am ok with many of them. I think some can be consolidated and some are not really necessary (Figure 19-20). Also, I am not sure what is going on with Figure 14. Is this in error? This figure legend does not match the presented figure in what I downloaded.

Another question I have is in relation to Figure 2. If the authors have data concerning the leaf loss in time per plot, could they potentially calculate a leaf decay rate (k). This would be useful for comparing to other macrophyte leaf species in aquatic habitats.

In conclusion, the authors have some interesting data on biotic and abiotic factors leading to damage and decay of floating leaved macrophytes. I feel this study was a lot of work for the authors and that it can contribute positively to our understanding of floating leaf macrophyte decay. However, this manuscript needs a serious and comprehensive editing before it can be adequately evaluated.

---

## Round 0.2 · Major Revisions

Thank you for your revised manuscript, which both previous referees agreed to review again. Unfortunately both conclude that the manuscript still requires very serious improvements and I side with them. Although they acknowledge the changes you made, both assessments concur that these are largely insufficient. The text still lacks clarity and an overall coherent flow of thought. In some instances, paragraphs still read like a collection of random sentences put together. In his/her first review, Referee 2 had pointed out that s/he illustrated some examples of passages that were symptomatic of the limitations of the manuscript. Both referees recognize that you specifically addressed the examples that they previously brought to your attention. However, it appears you did not realize that these were just examples of many problems throughout the manuscript. Thus, I hope you will be able and willing to revise the manuscript much more fundamentally, not just the passages the referees highlight this time.

The decision letter above is a standard text that I am unable to change. Therefore, please note that I cannot currently judge chances of acceptance or your manuscript, in contrast to the statement made in the letter. Instead, I would like to emphasize again that the manuscript will have to be improved very substantially during the next revision, because I am afraid another round of major revisions will not be possible.

·

Basic reporting

1.It is a little bit odd to have the explanation of the several marks found on the leaves in section Results. Typically, Results is a very straight forward section where no discussion or explanations are given; explanations, and the references used to support the explanations, would fit better the discussion than the results section;
2.Because some of the contents of the Discussion are in the Results, the Discussion -although greatly improved – still needs a little bit more work. In fact, there are only two references in the Discussion, which still is more a synthesis of the results than a discussion.
3.The discussion would also benefit from a reorganization of subjects. As it is, it goes back and forth from one issue to another, and it is difficult to follow by the reader.

Experimental design

no comment

Validity of the findings

no comment

Additional comments

The manuscript was greatly improved during the revision made by the authors: the written language was improved (I have some minor corrections/suggestions on the attached document); the text was rearranged, removing from materials and methods the sentences that clearly belonged to introduction; the reference list was shortened; the discussion was improved; the number of figures was reduced, and the raw data is now complete.
However, some of the problems reported by both reviewers still persist and thus the manuscript needs to suffer another round of revision before being accepted for publication in PeerJ.

Reviewer 2 ·

Basic reporting

See General comments to authors below.

Experimental design

As I stated before - The study is original and within the aims and scope of the journal. Overall experimental design seems sound and will fill a gap in knowledge concerning floating leaf macrophyte damage and decomposition. However, as stated in the general comments to authors below, I feel the manuscript is in need of editing to provide greater clarity

Validity of the findings

See General comments to authors below.

Additional comments

“Initial decomposition of floating leaf blades of waterlilies: causes, damage types and their impact” by Klok and van der Velde.

This revised manuscript version presents an overview of a series of field studies that investigated the biotic and abiotic factors involved in the initial decomposition of floating leaf blades of Nuphar lutea, Nymphaea alba, and Nymphaea candida. Here, floating leaf blades of N. lutea, N. alba, and N. candida were examined at three freshwater sites that differed in environmental conditions (trophic status, pH and alkalinity). Floating leaf blades were tagged/numbered within replicate plots and leaf length, percentages and types of damage to, and decay of, each leaf was measured and estimated weekly for all plots during the growing season. The manuscript describes the initial decomposition stages of leaves and the various causes (biotic and abiotic) that are chiefly responsible.

Overall Comment
As I stated in my prior review, I found that this manuscript to be interesting. It contains meticulous data on the types of damage to, and decay of, the leaf blades of water lilies, which the authors are well known for studying. It is a large data set that they have compiled and I applaud them for their efforts. Having that said, I admit that am rather disappointed in the present revision. In my prior review I stated… “My main concern with this paper is with the overall writing and logical flow of the manuscript. In my opinion, this manuscript is just not ready for publication and will require a very careful, lengthy revision.”

I still have this same opinion after reading this revised version. A large part of the writing within this manuscript needs editing for both clarity and flow. There are several confusing sentences and passages throughout the manuscript that are not well written or connected. As I mentioned in my prior review…“I would have so many specific comments concerning the text that it would be just pointless to list them all.”

In my prior review, I tried to specifically illustrate some examples of passages that were, in my opinion, symptomatic of the manuscript. While the authors specifically addressed these examples, they failed to realize or ignored that these were just some examples of many passages I encountered throughout the manuscript. It appears that they did just a casual editing (addressing what I only specifically listed), rather than engage in a serious polishing of the manuscript text.

Again, I will give just a FEW EXAMPLES in my specific comments below:

Abstract – Again, in my opinion the abstract needs to be restructured. I am ok with the first paragraph. However, the second paragraph needs to be entirely rewritten for the following reason:

The authors state - “Initial decomposition and its various causes are depicted and described. Also leaf damage with respect to the potential leaf area per species per plot, contributions to leaf damage by external causes, leaf loss in time and succession of damage causes are presented.”
Please do not tell me in the abstract of the manuscript that data will be depicted, described or presented within the manuscript. Please present a synopsis of the actual data in the manuscript within the abstract. In addition, the second paragraph of the abstract is an example of what I mean when a passage is not very well connected in terms of flow of thought. In my opinion, it reads like a collection of sentences thrown together. Please think about the overall importance of the study and the results (i.e., the take-home message) and fashion an appropriate abstract.


Introduction- first paragraph

Original - “Already during their development plant leaves are exposed to abiotic factors (such as weather conditions, causing physical damage, fragmentation and drying out) as well as biotic factors (such as infection by fungi and viruses, herbivores, and animals using these parts of the plant in various ways). This exposure is well-known for crops and ornamental plants as it causes economic damage.”

Revised – “Plant leaves are, already during their development exposed to abiotic factors such as
weather conditions, as well as biotic factors such as infection by fungi and viruses, herbivores, and animals using these leaves in various ways. This exposure is well-known for crops and ornamental plants as it causes economic damage.”

In my opinion, I find that instead thinking about and revising the first paragraph, the authors are simply involved in an exercise of moving words around. The new revised sentence above is even more awkward than the former.

These are just two examples, but there are several more scattered throughout the manuscript.

For example,

Line 76-79. These phases of initial decomposition can be studied well in the leaf blades (laminae) of large leaved plants such as waterlilies in which the turnover of floating leaf blades, further indicated in this paper as floating leaves or leaves, is low (P/Bmax 1.35-2.25) and the leaves exist for a relatively long time, on average 38-48 days (Klok & Van der Velde, 2017).

I really don know what the authors are trying to convey here?

Line 168-170. Is this passage really needed?

Line 184. Is Ci a correlation coefficient or a regression coefficient from the quadratic regression equation modeling leaf area from leaf length. Note the difference between correlation and regression. Please provide more details on this equation.

Line 216-217. “Initial decomposition tends to increase in time during the vegetation season with the exception of the acid Voorste Goorven (with Nuphar lutea and Nymphaea alba).” I have no idea what this sentence is trying to convey. To me, of course initial decomposition of leaves tends to increase with time as leaves grow, senesce and decay throughout vegetation season. In regards to Voorste Goorven, do the authors suggest here that initial decomposition does not proceed until the end of the growing season?

Line 217- 219 In the alkaline waters (Oude Waal and Haarsteegse Wiel) leaf damage for Nuphar lutea was less than 20% related to the total potential leaf area in the plot until half September…”
Half Semtember? How about mid-September.

Line 240-243. “Autolysis starts shortly after the first leaves are fully grown and continues the
whole floating leaf vegetation period. The leaf turns from green to yellow, which leads at
the end of the existence of the floating leaf to total microbial decay, the leaf turning brown.”

What do these two sentences mean? In addition, this passage is a good example of what I mean by a confusing and not well written/connected passage. The passage (Autolysis) starts out with… “The newly unrolled floating leaves are green and hydrophobic by an epicuticular wax layer (Fig. 4). During senescence this wax layer erodes by colonization of bacteria and fungi. In this stage the leaf tissue can be attacked by cellulolytic bacteria (Howard-Williams et al., 1978; Robb et al., 1979; Rogers & Breen, 1981; Barnabas, 1992).” What do the first two sentences have to do with autolysis? The authors don’t even begin to address autolysis until the third sentence? Are the above processes a result of autolysis?

“Autolysis is controlled by the plant itself by hormones (e.g. Osborne, 1963)”? What?

Line 245-248. “In October the percentage of affected leaves rose to 100%, however, the surface
area affected was quite stable and generally around 10%. For separate leaves the area affected by autolysis may decrease in time, since microbial decay will take over part of the area (Fig. 5).” I really have no idea what this sentence is trying to convey to the reader?

Line 255-257. “Due to hard wind floating leaves are lifted from the water, flip over and
subsequently air exposed parts (in particular the leaf margin) dry out (Fig. 8). The effect
on leaf surface area was generally less than 5%”. Lets use some commas – a example.

How about…Due to hard wind, floating leaves are often lifted from the water and flip over. Subsequently, leaves are exposed to air (in particular the leaf margin), leading to leaf dessication (Fig. 8). Overall, the effect of dessication stress on leaf surface area was generally less than 5%.

Line 259-261. The percentage of affected leaves was quite high during the whole data taking period for plots (Nuphar lutea, HW), (Nymphaea alba, OW) and (Nymphaea candida, HW), ranging about 60-80%. Awkward phrasing

Line 273- 275. Damage by consumption of leaf tissue by the Coot (Fulica atra) can be recognized by omissions in the form of triangular areas at the edge of a leaf. Omissions? What does this mean in relation to this sentence? Again, awkward sentence phrasing.

Line 290-292. “The imagines live on the floating leaf upper side where they feed on leaf tissue (upper epidermis, parenchym till the under epidermis).” Imagines? That one is new to me. I had to look it up. Although correct, it is a confusing term that most readers will find awkward, like “vital”, “sapropelium”, and “percentual”. Why not just state here and elsewhere “adults”, like “The adults live on the floating leaf upper side where they feed on leaf tissue (upper epidermis, parenchym till the under epidermis).”

Line 343-344. “With translucent light it appeared that the real damage was higher due to leakage, etc. What? This sentence is just sort of thrown in at the end of this paragraph with no context to the preceeding sentences.

The discussion, in my opinion, is in need of serious editing. This is yet another example of several paragraphs that read like a collection of random sentences (thoughts), that are not very well connected in terms of flow of thought. In addition, some of what is presented in the discussion is just a reiteration of what is in the results and thus is rather redundant. Dido for the conclusion.

In conclusion, the authors have some interesting data on biotic and abiotic factors leading to damage and decay of floating leaved macrophytes. I feel this study was a lot of work for the authors and that it can contribute positively to our understanding of floating leaf macrophyte decay. However, this manuscript needs a serious and comprehensive editing. I just cannot go through this manuscript and provide a specific example and justification for everything. I would literally be going sentence by sentence throughout the entire manuscript. I hate to be so negative about this revision. I was hoping the authors would make a diligent attempt at comprehensive revision, as I felt, and still feel that the data presented in this manuscript is interesting and beneficial to the scientific community. I now urge the authors to spend more time editing this manuscript. Ultimately the authors greatly diminish this work and its potential impact by the present manuscript text.

---

## Round 0.3 · Major Revisions

We are in a very unusual situation that your manuscript has now been revised twice and reviewed three times, but although both reviewers acknowledge that the manuscript has notably improved during both rounds of revisions, it is still not ready for publication. As you will notice in the attached comments, both reviewers would like to see your paper in print, but also consider that more work is needed to bring it to that level. However, as you will also notice, they lost steam in making detailed specific suggestions. Therefore, I would like to ask you to revise the manuscript again not just by responding to the few specific comments of Reviewer #1, but also to further improve the overall structure, as pointed out particularly by Reviewer #2. Put differently, please make another systematic attempt to edit and polish the textual presentation.

·

Basic reporting

The manuscript was greatly improved during the second revision made by the authors, which tried to follow all suggestions made by both referees. There are still a few minor language problems which I tried to fix in the attached annotated manuscript. But at least two sentences have to be completely re-written in order to make them understandable for the reader. I rephrased them in order to help the authors, but since they were difficult to understand I am not sure if the meaning I gave them was the meaning intended by the authors.
Some of the aspects needing attention include:
Lines 190-194 & 292-293: Delete the sentence marked in blue. In the first sentence you state that the results of translucent light were not going to be reported. If they are not being reported than you don’t explain the method. In the second sentence you report a result with translucent light, which in fact is not necessary.
Lines 336-338: Very difficult to follow. I suggest rephrasing for something like “Leaves can grow during 53 to 73% of the vegetation period (citations). In the present study the development of new leaves and the dying of old ones continued for as long as... (state a period instead of saying "a long period")
Lines 369-372: rephrase sentence marked in blue. Maybe “Cricotopus trifasciatus, which causes damage to the floating leaves of Nymphoides peltata (Gmel.) O. Kuntze (Lammens & Van der Velde, 1978) was also observed to cause damage on waterlily leaves (Van der Velde & Hiddink, 1987).” Although is still confusing because of the general term “waterlily”. Maybe state the species instead of using the word waterlilies?
Lines 390-394. Very confusing sentence. Rephrase. Maybe “"Nymphaea candida (HW) showed an increase in nail scratches towards the end of June, which may be the influence of young coots. In fact, high densities of waterfowl lead to higher damage of the leaves, and the meadows surrounding OW and HW important for coot to survive winter time by grazing grass in groups. “
Lines 398-404: This was really difficult to follow. If I understood correctly, maybe the sentence could be re-written as “The inhibition of cell wall degradation slows leaf fragmentation and leads to increased storage of phenolics, decreasing fragmentation and preventing the softening of the tissue. Thus the plant tissue constitutes a low quality resource for detritivores (Kok, 1993) which are also inhibited by high Al concentrations and low pH.” (and the next sentence which is on other subject should be in the beginning of the next paragraph)

Experimental design

Not applicable on this round of revision

Validity of the findings

Not applicable on this round of revision

Reviewer 2 ·

Basic reporting

See General comments to authors below.

Experimental design

The study is original and within the aims and scope of the journal. Overall experimental design seems sound and will fill a gap in knowledge concerning floating leaf macrophyte damage and decomposition. However, as stated in the general comments to the editor and authors (see below), I feel the manuscript is still in need of editing to provide greater
clarity.

Validity of the findings

See General comments to authors below.

Additional comments

“Initial decomposition of floating leaf blades of waterlilies: causes, damage types and their impact” by Klok and van der Velde.

This revised manuscript (2nd) version presents an overview of a series of field studies that investigated the biotic and abiotic factors involved in the initial decomposition of floating leaf blades of Nuphar lutea, Nymphaea alba, and Nymphaea candida. Here, floating leaf blades of N. lutea, N. alba, and N. candida were examined at three freshwater sites that differed in environmental conditions (trophic status, pH and alkalinity). Floating leaf blades were tagged/numbered within replicate plots and leaf length, percentages and types of damage to, and decay of, each leaf was measured and estimated weekly for all plots during the growing season. The manuscript describes the initial decomposition stages of leaves and the various causes (biotic and abiotic) that are chiefly responsible.

Overall Comment
As stated in my two prior reviews, I find this manuscript to be interesting. It contains meticulous data on the types of damage to, and decay of, the leaf blades of water lilies, which the authors are well known for studying. It is a large data set that they have compiled and I applaud them for their efforts. The present version has certainly improved from the first submission. However, I must admit that I do not see any systematic attempt by the authors to edit and polish the textual presentation of this manuscript. In my opinion, the present manuscript is only marginally ready. This is unfortunate as I believe the current presentation diminishes this work and its potential scientific impact.

At present, I am not sure what more I can add to this discussion apart from what I have tried to communicate in my two prior reviews. I really hate to be so negative about this manuscript. The authors have some interesting data and it can contribute greatly to our understanding. Unfortunately, I feel that the textual presentation is still in need of some restructuring for clarity and flow.

---

## Round 0.4 · Major Revisions

This time I have not forwarded your revised manuscript to the reviewers but instead worked on it in greater detail myself. As you will see in the attached files (main text, Tables 1 and 2), I have numerous editorial suggestions and several remaining queries, many of them along the lines of the criticism expressed by the reviewers.

Furthermore, I would like to ask you to present your results in a more quantitative fashion. Your sampling design, which involved 6 plots in each stand, and your quantitative measures make this possible, including statistical comparisons.

Please also pay greater attention to distinguishing results and interpretation.

All Figures and Tables need a legend.

Finally, I noted a number of terms used throughout the manuscript where the exact meaning in the context of this study did not become clear to me, partly because you do not explain what evidence was used to assign a particular type of leaf damage to these categories. These terms include “autolysis” (turning from green to yellow does not indicate autolysis but senescence), “fragmentation,” “microbial decay.” As a result, not all the changes called “initial decomposition” qualify as decomposition.

---

## Round 0.5 · Minor Revisions

Thank you for your thorough revision of the manuscript, which I believe has been notably improved. Nevertheless, I do have a few additional queries and suggestions, as you will see in the attached file. In particular, I am still unhappy with your definition of ‘autolysis’ and ‘microbial decay.’ Please reconsider (rename) them (in the ms, tables and figs) in the light of the comments I wrote in the margin of the manuscript. Mainly for this reason I would also like to ask you to remove Figure 1. Please also add captions to all other tables and figures.

I will attach PDF of the main manuscript and Tables 1-4. In addition, I will ask the staff of PeerJ to forward you the Word files I worked on because the PDFs do not show my notes in the margin.

I look forward to hearing from you in due course.

---

## Round 0.6 · Minor Revisions

I am sorry that I must return parts of the manuscript once again. For some reason, I had not seen the legends of figures and tables in previous versions, so I could only add my editorial comments on those now. You will find attached a file containing these changes in track-changes mode. I also attach in the same file once again the ms and Tables 3 and 4 with a few trivial final edits. I will ask to send you the corresponding Word files as well, especially because comments and changes are not all visible in the PDF file.

---

## Round 0.7 · accepted · Accept

Thank you for your patience.

#